# Antibiotic Resistance Awareness in Kosovo: Insights from the WHO Antibiotic Resistance: Multi-Country Public Awareness Survey

**DOI:** 10.3390/antibiotics14060599

**Published:** 2025-06-12

**Authors:** Flaka Pasha, Valon Krasniqi, Adelina Ismaili, Shaip Krasniqi, Elton Bahtiri, Hasime Qorraj Bytyqi, Valmira Kolshi Krasniqi, Blana Krasniqi

**Affiliations:** 1Department of Pharmacology and Toxicology and Clinical Pharmacology, Faculty of Medicine, University of Prishtina “Hasan Prishtina”, 10000 Prishtine, Kosovo; flaka.pasha@uni-pr.edu (F.P.); shaip.krasniqi@uni-pr.edu (S.K.); elton.bahtiri@uni-pr.edu (E.B.); hasime.qorraj@uni-pr.edu (H.Q.B.); 2Department of Clinical Pharmacology, University Clinical Center of Kosovo, 10000 Prishtine, Kosovo; 3Department of Nursing and Midwifery, Faculty of Medicine, University of Prishtina “Hasan Prishtina”, 10000 Prishtine, Kosovo; adelina.ismaili@uni-pr.edu; 4Department of Chemistry, Faculty of Natural and Mathematical Sciences, University of Prishtina “Hasan Prishtina”, 10000 Prishtine, Kosovo; v_mira_5@hotmail.com; 5Department of General Medicine, University of Medicine, 1009 Tirana, Albania; krasniqiblana@gmail.com

**Keywords:** antimicrobial resistance (AMR), antibiotic misuse, public health awareness, self-medication, One Health approach

## Abstract

**Background/Objectives:** Antimicrobial resistance (AMR) poses a critical global health threat, rendering common bacterial infections increasingly difficult to treat and placing considerable strain on healthcare systems. This study assesses public awareness, perceptions, and behaviors related to antibiotic use and AMR in Kosovo, a country with limited existing data on the topic. **Methods:** Using a cross-sectional survey design, 568 participants from diverse demographic backgrounds provided insights into their knowledge and practices concerning antibiotic use and antibiotic resistance. **Results:** The results revealed that although 75% of participants had heard of antibiotic resistance, only a limited proportion understood key terms. Knowledge of appropriate antibiotic use varied, with 67% of respondents correctly recognizing the need to complete a prescribed antibiotic course, while 29% believed it was acceptable to stop treatment once they felt better. Gender and educational level emerged as significant factors, with women and more educated individuals demonstrating greater awareness of proper antibiotic use and the risks of misuse. While 71% of respondents considered it unacceptable to use antibiotics prescribed to others, 41% believed it was acceptable to reuse previously effective antibiotics. Most participants (96%) reported obtaining antibiotics through prescriptions. Public awareness of AMR was generally high, but conceptual understanding remained limited, with misconceptions about the origins of resistance, incorrectly attributing it to the human body rather than bacteria. **Conclusions:** Targeted public health campaigns, guided by the One Health approach, integrating human, animal, and environmental health, are needed. A multifaceted strategy, including education, policy reforms, and international collaboration, is essential to mitigate AMR and preserve the efficacy of antibiotics for future generations.

## 1. Introduction

Antimicrobial resistance (AMR) is a growing global concern, with the potential to render common bacterial infections untreatable and significantly increase mortality rates. It represents a major threat to public health, as recognized by the World Health Organization [1]. Routine medical procedures, including cancer treatments, surgeries, and other essential healthcare interventions, are becoming riskier due to the evolution of infectious agents that are resistant to existing antibiotics, placing additional strain on healthcare systems [2]. The overuse and misuse of antibiotics in both medical and agricultural settings are key drivers of AMR, highlighting the urgent need for public awareness campaigns and educational programs to curb its spread [3].

AMR is no longer only a theoretical concern; rather, it is a present and escalating crisis with direct consequences for patient safety, health system sustainability, and economic stability. The latest global burden estimates suggest that antimicrobial resistance was associated with nearly 5 million deaths in 2019, with projections warning of up to 10 million deaths annually by 2050 if effective action is not taken [4,5]. Despite the urgency, public understanding of AMR remains uneven, with people being unaware of how their individual behaviors, such as antibiotic misuse or demanding antibiotics for viral infections, contribute further to the problem [6].

Beyond individual behavior, structural issues such as weak pharmaceutical regulation, limited access to diagnostic testing, and the availability of antibiotics without a doctor’s prescription contribute to the inappropriate use of antibiotics in many regions [7]. Therefore, solutions must be both population-specific and context-sensitive, grounded in a clear understanding of what people know, believe, and do regarding antibiotics. Knowledge, attitude, and practice (KAP) studies are essential tools in this regard, enabling public health authorities to design targeted interventions that address local misconceptions and behavioral patterns [8].

In addition to behavioral factors, there is a growing global emphasis on the “One Health” approach, which recognizes that the health of people is closely connected to the health of animals and the environment. Antibiotics used in agriculture and veterinary medicine can lead to the development of resistant bacterial strains, which may be transmitted to humans through food, water, and environmental pathways [9]. The interconnected nature of AMR underscores the need for integrated surveillance systems, public education that includes animal and environmental health perspectives, and cross-sectoral collaboration [10]. Despite its importance, the One Health dimension is often overlooked in public health campaigns, especially in countries with limited institutional coordination across sectors [11].

Kosovo, like many countries in the region, faces multiple challenges that make it particularly vulnerable to the spread of AMR. These include limited surveillance infrastructure, weak enforcement of pharmaceutical regulations, and widespread availability of antibiotics without a prescription, despite legal restrictions on over-the-counter sales [12]. Additionally, access to healthcare remains uneven, especially in rural areas, which may encourage self-medication and informal treatment practices [13]. Understanding public perceptions of AMR in Kosovo is, therefore, not only urgent but foundational for the development of national strategies that align with global AMR containment goals [14].

This study provides one of the first systematic assessments of public awareness, attitudes, and behaviors regarding antibiotic use and antimicrobial resistance in Kosovo. It fills a critical knowledge gap while also offering insight into how One Health-oriented education and policy reforms might be developed to enhance antibiotic stewardship in both human and animal health sectors [15].

## 2. Results

### 2.1. Participants’ Demographics

The sample consisted of 568 participants with diverse demographic backgrounds. Most were ethnic Albanians (96.6%), and almost 60% identified as male. Participants represented a range of age groups, educational levels, income brackets, and household compositions. A summary of key demographic characteristics is presented in Table 1.

### 2.2. Proper Antibiotic Use and AMR Awareness

The majority of participants (64%) reported having used antibiotics within the last year, and nearly one-third (32%) had used them within the past month (Figure 1).

Among those with prior antibiotic use (n = 530), almost all (95%) obtained antibiotics via a doctor’s prescription, and 92% reported receiving instructions on how to take them. A small minority (4%) did not consult a healthcare provider or did not receive or recall receiving guidance (Figure 2).

When asked about the appropriate antibiotic treatment duration, 67% of respondents correctly indicated that antibiotics should be taken until all doses are completed, while 29% believed treatment can stop when symptoms improve. Correct knowledge was significantly more common among women (74% vs. 60% of men) and participants with a Bachelor’s degree or higher (over 80%) (Figure 3).

Logistic regression confirmed that female gender (OR = 2.05, 95% CI [1.40, 3.01], *p* < 0.001) and higher education (Bachelor’s or higher vs. ≤high school: OR = 2.87, 95% CI [1.89, 4.36], *p* < 0.001) independently predicted correct responses.

Regarding the acceptability of using antibiotics prescribed to a friend or family member, 71% correctly indicated that this practice is inappropriate. However, 25% believed it was acceptable if the illness appeared similar, and 4% were unsure. Women (77%) and more educated individuals were again significantly more likely to respond correctly (Figure 4). In logistic regression models, both female gender (OR = 1.81, 95% CI [1.23, 2.67], *p* = 0.003) and higher education (OR = 2.45, 95% CI [1.60, 3.76], *p* < 0.001) were associated with correct knowledge.

When asked whether it is acceptable to request or reuse antibiotics that previously worked for a similar illness, 41% incorrectly agreed with this practice, while 55% disagreed. This misconception was significantly less common among participants with postgraduate education, where only 18% endorsed it (Figure 5).

Logistic regression showed that having a Master’s degree or higher was associated with significantly lower odds of endorsing this misconception (OR = 0.28, 95% CI [0.14, 0.56], *p* < 0.001).

To further examine patterns in knowledge and attitudes toward antibiotic use, we conducted a series of logistic regression analyses using key demographic predictors. Specifically, we modeled the likelihood of respondents providing correct responses to three knowledge items corresponding to Figure 3, Figure 4 and Figure 5. Independent variables included gender and education level, with all models adjusted for age and income. The outcomes were binary, coded as 1 for a correct or appropriate response and 0 for incorrect or inappropriate responses. Odds ratios (ORs), 95% confidence intervals (CIs), and *p*-values are reported in Table 2.

A substantial majority of respondents (76%) agreed that antibiotic resistance is one of the most serious global health challenges, and 88% believed that individuals must use antibiotics responsibly. Most participants also expressed confidence that healthcare professionals will be able to manage AMR. However, a lower proportion strongly agreed with incentivizing pharmaceutical companies and governments to develop new antibiotics (54% and 56%, respectively).

When asked about antibiotic use in food-producing animals, 61% of respondents reported being unsure. One-quarter (25%) believed antibiotics are widely used in agriculture in Kosovo, while 13% believed they are not.

## 3. Discussion

This study provides valuable insight into general knowledge, attitudes, and practices regarding antibiotic use and antimicrobial resistance in Kosovo, a country where data on the subject remain quite limited. The survey reveals both encouraging results and concerning gaps in understanding, particularly among certain demographic groups. These findings underscore the need for targeted public health interventions and sustained investment in antibiotic stewardship efforts.

The demographic profile of the sample, although not nationally representative, offers a meaningful cross-section of the population, dominated by ethnic Albanians and skewed toward urban areas. The predominance of respondents aged 25–34 suggests a young and potentially more digitally literate population, which may influence both healthcare-seeking behavior and exposure to public health campaigns. Notably, the survey shows a relatively high level of educational attainment among participants, a factor that has consistently been associated with greater health literacy, including awareness of appropriate antibiotic use [16]. Compared to national data, the study sample included a higher proportion of individuals with tertiary education and a slightly younger age distribution.

According to the Kosovo Agency of Statistics, approximately 16% of the general population holds a university degree, whereas over 30% of our sample reported a Bachelor’s degree or higher. Similarly, individuals aged 25–34 were overrepresented in this study, likely reflecting the greater accessibility and use of healthcare services in this age group. While this demographic profile may limit generalizability, it also offers valuable insight into groups most likely to engage with health information and behavior change campaigns.

However, it is important to acknowledge potential sampling bias. The sample was skewed toward urban residents and individuals with higher educational attainment, which may not fully reflect the broader Kosovo population, particularly those in rural areas with more limited access to healthcare and health information. These differences could result in overestimates of awareness and knowledge levels in the general population. Future studies should aim to recruit more representative samples across geographic and socioeconomic strata to enhance generalizability and inform broader intervention strategies.

A central finding of this study is the widespread use of antibiotics, with 32% of participants reporting use within the past month. Although a majority acquired antibiotics via prescription (95%), the frequency of recent use raises concerns about possible over-prescription or self-diagnosed treatment of minor illnesses. High rates of antibiotic consumption, even when medically supervised, have been linked to the development of resistant bacterial strains, particularly in low- and middle-income countries [17]. The implications of such patterns are especially critical in healthcare systems with limited laboratory capacity to guide appropriate therapy. In Kosovo, the sale of antibiotics without a prescription is legally prohibited. According to national legislation and administrative guidelines, pharmacists are authorized to dispense antibiotics only upon the presentation of a valid prescription. However, despite these regulations, enforcement remains a significant challenge. Investigations have revealed that antibiotics are frequently sold over the counter without prescriptions in many pharmacies across Kosovo. This practice is often attributed to factors such as insufficient regulatory oversight, a limited number of pharmaceutical inspectors, and a general lack of awareness among the public regarding the risks associated with improper antibiotic use. The widespread availability of antibiotics without proper medical guidance contributes to the misuse of these medications, which in turn accelerates the development of antimicrobial resistance (AMR). This situation underscores the urgent need for stricter enforcement of existing laws, increased public education on the responsible use of antibiotics, and the implementation of comprehensive strategies to combat AMR in Kosovo.

Knowledge of appropriate antibiotic use was diverse. While most participants recognized the importance of completing the full course of antibiotics, nearly 29% indicated that they would stop taking antibiotics once they felt better. This misunderstanding aligns with global trends that continue to challenge public health efforts despite long-standing WHO guidance. The premature discontinuation of antibiotics can lead to partial suppression of infection and selection for resistant organisms, undermining treatment efficacy and increasing the burden on healthcare systems [18]. In real-world terms, this behavior may result in prolonged illness, increased risk of complications, and the transmission of resistant pathogens within the community. For health systems with limited diagnostic and follow-up capacity, as in Kosovo, these outcomes place additional pressure on already resource-constrained services and may contribute to avoidable hospitalizations and antibiotic escalation [19].

Gender and education were notable factors in antibiotic literacy. Women were significantly more likely than men to correctly understand the importance of completing prescribed treatments. This gender difference may reflect broader trends in health-seeking behavior, with women often more engaged in preventive health practices and more likely to consult healthcare providers [20]. Likewise, respondents with higher levels of formal education demonstrated significantly better knowledge on appropriate antibiotic use and the risks of self-medication [21]. These associations highlight the need for tailored education strategies that consider social and educational disparities.

Public awareness of antibiotic resistance was relatively high, with 75% of respondents having heard of the term. However, a deeper conceptual understanding was lacking. An overwhelming majority (99%) incorrectly believed that antibiotic resistance arises in the human body rather than in bacteria, a misconception that mirrors findings from studies in other countries [22]. Such misunderstandings can diminish the perceived urgency of antibiotic resistance as a societal threat and impede behavior change. Most respondents reported learning about AMR through healthcare professionals, suggesting that while clinical settings are an important channel for public education, the messaging may not always be effective in conveying scientific concepts.

Further complicating the issue is the widespread belief that antibiotics are effective against a range of conditions, including sore throats, skin infections, and urinary tract infections. While antibiotics may be appropriate for some bacterial infections, they are often misused for viral conditions where they offer no clinical benefit. Similar trends have been reported globally, contributing to unnecessary prescribing and rising resistance rates. These data highlight the importance of differentiating between bacterial and viral infections in public health messaging and clinical consultations [23].

One of the less explored but increasingly critical aspects of AMR is the use of antibiotics in agriculture. A substantial portion of respondents (61%) were unsure whether antibiotics are used in Kosovo’s farming sector, and only 25% believed such use was widespread. This uncertainty indicates a lack of public awareness regarding the interconnectedness of human, animal, and environmental health, a core principle of the One Health approach. The role of agricultural antibiotic use in accelerating resistance has been well documented, particularly in livestock production, where sub-therapeutic dosing is commonly employed to promote growth and prevent disease [24]. Public education campaigns must therefore broaden their scope to include food safety and agricultural practices as part of comprehensive AMR prevention strategies.

This study comes with a limitation. The reliance on self-reported data may be subject to recall or social desirability bias. Nevertheless, the findings offer valuable insights into prevailing knowledge and behaviors regarding antibiotic use and resistance in Kosovo and highlight critical areas for intervention.

## 4. Materials and Methods

### 4.1. Study Design

A cross-sectional survey design was employed to explore public knowledge, attitudes, and practices regarding antibiotics, antibiotic resistance, and their use in healthcare and agriculture within Kosovo. This study was quantitative in nature, utilizing structured questionnaires to gather data on demographic characteristics, antibiotic usage patterns, knowledge of antibiotics, and attitudes towards antibiotic resistance.

### 4.2. Study Participants

The sample consisted of 568 participants, who registered and received primary care services as patients. This sample represented a diverse demographic cross-section of the population in Kosovo, considering factors such as ethnicity, gender, age, educational level, and geographic location (urban, suburban, rural).

### 4.3. Participant Recruitment and Sample Size

Participants were recruited using a convenience sampling strategy across multiple primary healthcare centers in Pristina. Inclusion criteria were having the capacity to provide informed consent and being able to complete the survey in Albanian. The survey was administered in person by trained researchers during regular clinic hours. Of the approximately 750 individuals approached, 568 agreed to participate, resulting in a response rate of 75.7%. A formal sample size calculation was not conducted prior to data collection. However, a minimum sample of 384 participants was estimated post hoc using a 95% confidence level, 5% margin of error, and an assumed 50% proportion for knowledge of antibiotic resistance in the population. Our final sample of 568 exceeds this threshold, allowing for subgroup analyses by gender and education level.

### 4.4. Data Collection

Data were collected through the Antibiotic Resistance: Multi-Country Public Awareness Survey [25]. We obtained the required permission to use the “Antibiotic Resistance: Multi-Country Public Awareness Survey” developed by the World Health Organization (WHO). Additionally, we translated the survey into Albanian in accordance with the necessary guidelines and permissions. The questionnaire collected a broad range of information from participants, including demographic details such as ethnicity, gender, age, education, income, and household structure. Additionally, the survey explored patterns of antibiotic use, including whether individuals consulted healthcare professionals and received instructions on proper usage. The survey assessed participants’ knowledge of appropriate antibiotic use and their understanding of issues related to antibiotic resistance, including familiarity with terms like “superbugs” and sources of information on the topic. It also examined public attitudes toward the severity of antimicrobial resistance and potential strategies for addressing it. Additionally, participants shared their perceptions regarding the use of antibiotics in the agricultural sector in Kosovo.

### 4.5. Data Analysis

Descriptive statistics were used to summarize demographic data and participants’ responses. Percentages were calculated for each survey item to provide an overview of behaviors and knowledge related to antibiotics. Chi-square tests were conducted to assess associations between categorical variables, particularly with respect to gender and education level. Significant patterns in knowledge and attitudes towards antibiotic use were examined across these demographic variables (*p* < 0.01). Data entry, cleaning, and analysis were performed using SPSS statistical software v.27, ensuring the efficient handling of large datasets and the accuracy of the results.

### 4.6. Ethical Considerations

This study adhered to established ethical guidelines, including human participants. Participants were informed of this study’s objectives, and informed consent was obtained before participation. All responses were anonymized to ensure confidentiality. Ethical approval was granted by the Main Primary Medical Center of Prishtina Institutional Review Board (protocol no. 2504, date: 12 July 2018).

## 5. Conclusions

In conclusion, while there is a relatively strong foundation of awareness surrounding antibiotic use in Kosovo, substantial knowledge gaps remain, particularly in understanding resistance mechanisms and the risks associated with self-medication and incomplete treatment. These gaps appear to be influenced by demographic variables such as age, gender, and education level. Public health strategies must prioritize accessible, culturally appropriate educational interventions that address these discrepancies. Furthermore, the role of healthcare professionals as trusted sources of information should be strengthened through continued training in risk communication. Our findings also indicate that public understanding of the role of antibiotics in agriculture and the broader One Health perspective remains limited. Therefore, greater efforts are needed to raise awareness about how antibiotic use in animals and the environment contributes to antimicrobial resistance.

Based on these findings, we recommend the implementation of a national AMR awareness campaign that includes school-based education, targeted messaging for high-risk groups (e.g., young adults, individuals with lower education), and stricter enforcement of prescription-only antibiotic policies in community pharmacies. Training programs for healthcare providers should be expanded to improve their capacity to educate patients effectively and discourage inappropriate antibiotic use.

At the policy level, Kosovo should strengthen its regulatory framework to monitor and limit the sale of antibiotics without a prescription, expand pharmacovigilance systems, and develop public education materials aligned with WHO guidelines.

In alignment with global AMR action plans, Kosovo must adopt a multidisciplinary, One Health framework that integrates human, animal, and environmental health to curb the rise in antibiotic resistance and ensure sustainable healthcare outcomes.

## Figures and Tables

**Figure 1 antibiotics-14-00599-f001:**
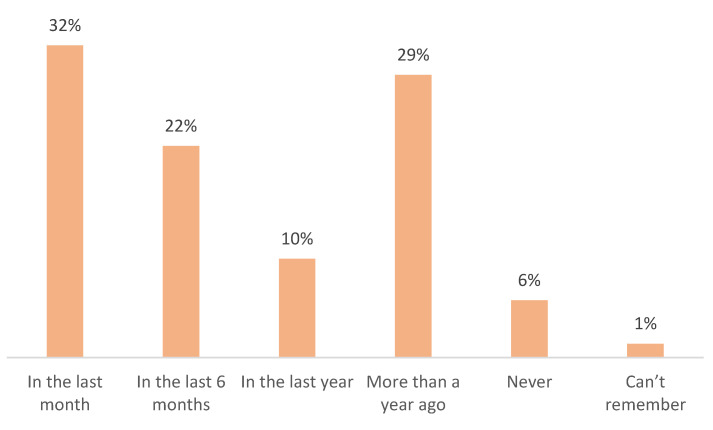
Timing of last antibiotic use among survey respondents (N = 568).

**Figure 2 antibiotics-14-00599-f002:**
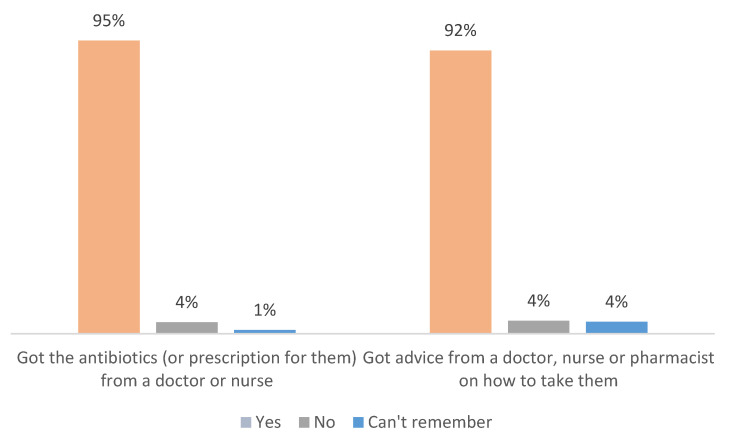
Rates of professional consultation and instructions received before or during antibiotic use (n = 530).

**Figure 3 antibiotics-14-00599-f003:**
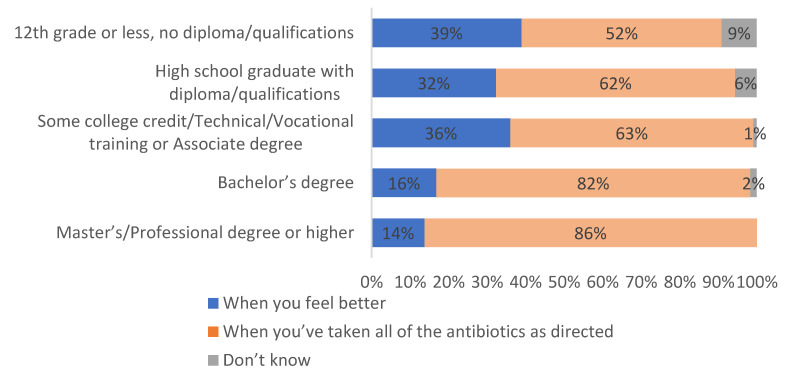
Perceptions of appropriate antibiotic treatment duration by gender and education (N = 568).

**Figure 4 antibiotics-14-00599-f004:**
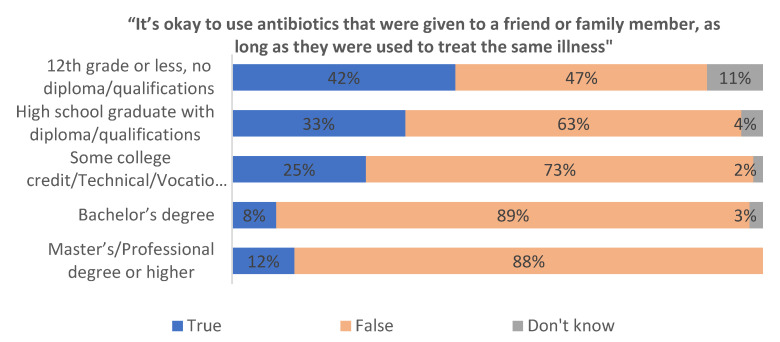
Beliefs about using antibiotics prescribed to friends or family members (N = 568).

**Figure 5 antibiotics-14-00599-f005:**
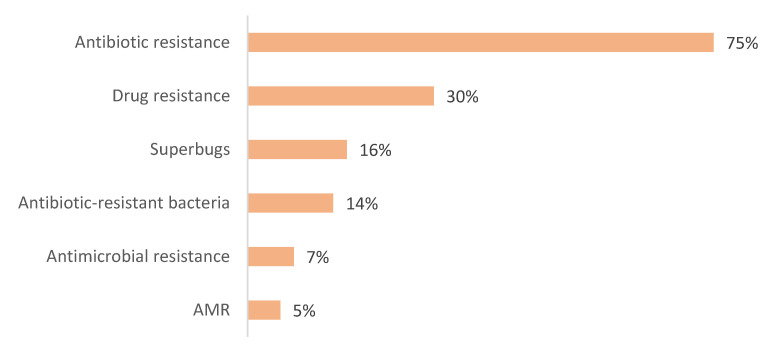
Beliefs about reusing previously effective antibiotics (N = 568).

**Table 1 antibiotics-14-00599-t001:** Summary of participant demographic characteristics.

Characteristic	Categories	*n* (%)
Ethnicity	Albanian, Other	549 (96.6%), 19 (3.4%)
Gender	Male, Female	340 (59.9%), 228 (40.1%)
Residence	Urban, Suburban, Rural	340 (59.9%), 179 (31.5%), 49 (8.6%)
Age Group	16–18, 19–24, 25–34, 35–44, 45–54, 55–64, 65+	47 (8.2%), 86 (15.1%), 144 (25.4%), 115 (20.2%), 106 (18.6%), 53 (9.4%), 18 (3.2%)
Education Level	No schooling, <HS, HS, Some college, Associate, Bachelor, Master+, Doctorate	22 (3.9%), 55 (9.7%), 206 (36.2%), 86 (15.2%), 27 (4.8%), 120 (21.2%), 48 (8.5%), 3 (0.5%)
Monthly Income (EUR)	<170, 170–250, 250–350, 350–650, 650–850, 850–1000, >1000	47 (8.2%), 57 (10.0%), 122 (21.5%), 156 (27.5%), 75 (13.3%), 34 (6.0%), 77 (13.6%)
Household Type	Multi-adult w/o children, w/children; Married/domestic w/or w/o children; Single, Single w/children	153 (27.0%), 137 (24.2%), 89 (15.6%), 74 (13.0%), 77 (13.6%), 38 (6.7%)

**Table 2 antibiotics-14-00599-t002:** Logistic regression results for knowledge and attitudes related to antibiotic use (N = 568).

Outcome	Predictor	OR	95% CI	*p*-Value
Correct belief: Complete full course of antibiotics (Figure 3)	Female (vs. Male)	2.05	[1.40, 3.01]	<0.001
	Bachelor’s+ (vs. ≤High School)	2.87	[1.89, 4.36]	<0.001
Correct belief: Do not use antibiotics prescribed to others (Figure 4)	Female (vs. Male)	1.81	[1.23, 2.67]	0.003
	Bachelor’s+ (vs. ≤High School)	2.45	[1.60, 3.76]	<0.001
Incorrect belief: Reuse previously effective antibiotics (Figure 5)	Master’s+ (vs. ≤High School)	0.28	[0.14, 0.56]	<0.001

## Data Availability

The original contributions presented in this study are included in the article. Further inquiries can be directed to the corresponding author.

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
