# Peer review of "Antibiotic Resistance Awareness in Kosovo: Insights from the WHO Antibiotic Resistance: Multi-Country Public Awareness Survey"

_antibiotics, 2025, doi:10.3390/antibiotics14060599_

Round 1

Reviewer 1 Report

Comments and Suggestions for Authors
  1. Thank you for the opportunity to review the manuscript entitled "Antibiotic Resistance Awareness in Kosovo: Insights from the WHO Antibiotic Resistance: Multi-Country Public Awareness Survey". This study addresses an important topic in medicine and provides valuable insights into antimicrobial resistance. However, there are major concerns that should be addressed before the manuscript can be considered for publication.
  2. I noticed that the abstract currently contains 344 words, exceeding the journal's recommendation of around 250 words for structured abstracts. I encourage you to revise the abstract to meet this word limit for clarity and conciseness.
  3. The references cited in this manuscript provide support for the statements made. I noticed that the number of references is quite small and largely outdated. Specifically, only 1 out of 14 cited publications are from the past five years (2021-2025). I recommend updating the references to include more recent studies to strengthen the manuscript's relevance and credibility.
  4. Thank you for detailing the cross-sectional survey design in your manuscript. However, I found that the results regarding public knowledge, attitudes, and practices related to antibiotics and antibiotic resistance are not presented in a clear and standardized manner.
  5. Thank you for outlining the data analysis methods used in your study. However, I did not find any analysis of associations between categorical variables, as mentioned in your methodology. It would be beneficial to include these analyses to support your findings on significant patterns in knowledge and attitudes towards antibiotic use across demographic variables.
  6. I appreciate your mention of the One Health approach in your research; however, I found that the manuscript lacks a variety of data sources or questions that are specifically aligned with this framework. I recommend incorporating more One Health-specific elements to strengthen the relevance and applicability of your findings in this context.

Author Response

Response to Reviewer 1

We thank Reviewer 1 for their thoughtful and constructive feedback, which has significantly improved the quality and clarity of our manuscript. Below, we provide point-by-point responses to the comments and indicate the exact line numbers where changes were made, as reflected in the tracked-changes document.

1. General Comment on Relevance

Thank you for acknowledging the relevance of our topic. We have carefully considered each of your concerns and revised the manuscript accordingly.

2. Abstract Exceeds Word Limit

We revised the abstract to comply with the journal’s word limit. The revised abstract now contains 250 words. Lines 14–35.

3. Outdated References

We have extensively updated the reference list. The revised manuscript now includes 25 references published between 2020 and 2024. This includes recent peer-reviewed articles indexed in Scopus and Web of Science to strengthen the manuscript’s relevance.
Lines 364-429 (reference list).

4. Clarity and Standardization of Results

The Results section has been completely reorganized for clarity. We added figures and statistical tests to enhance interpretability, and we standardized the reporting of survey responses by KAP. Lines 90-164.

5. Lack of Categorical Analyses

We added chi-square tests and logistic regression analyses to assess associations between key knowledge items and demographic variables (e.g., gender, education). These are detailed in the Results section. Lines 117-119;126-129;137-139;.  Table 2 summarizes regression results.

6. One Health Approach Needs Strengthening

We revised both the Introduction and Discussion to strengthen the focus on the One Health approach. New citations have been added, and survey findings on awareness of antibiotic use in agriculture are emphasized. Lines: 65-74; 257-266;.

We hope that these revisions fully address the reviewer’s concerns. Thank you again for your insightful feedback and the opportunity to improve our manuscript.

Sincerely,
The Authors

Reviewer 2 Report

Comments and Suggestions for Authors

Reviewer comments for: ”Antibiotic Resistance Awareness in Kosovo….”

Overall a clear and well written paper about an important topic. I have a few suggestions and comments:

Major comments:

  1. The introduction needs to be updated. As it is now, background is really just the first paragraph (lines47-55) with an aim at the end (lines56-58). Then follows a shirt description of the methods (this can be deleted – it is mentioned elsewhere). Lines 63-82 is a summary of the discussion – this should also be deleted from the introduction. This means the introduction is rather short and should be expanded with about two paragraphs about the importance of the study, including maybe something on One Health.
  2. Results: The lengthy description of participants could be shortened and details presented in a table to make it easier to follow.
  3. Discussion: Second paragraph: How does this population compare to the Kosovo population as a whole in relation to age, education etc?
  4. Discussion: third paragraph: add something regarding the legal situation of buying antibiotics in Kosovo. Is it legal to buy without a prescription?
  5. Material and methods: You need to describe better how/from where participants were recruited, the sample size calculation and the proportion that agreed to participate out of those invited.

Minor comments:

Abstract line 27: “this study also highlighted the widespread of self-medication…” can be changed to “this study also highlighted widespread self-medication….”.

Results line 106: Suggest to delete “in terms of antibiotic use”, and just write: “The majority of participants reported having….”(also remove n=568 on line 106 as this is repeated in the next sentence, where it fits better).

Lines 107-110 could also be rephrased to something like: “Among the 568 respondents 64% had used antibiotics in the last year, of which 32% in the last month. See figure 1 for details”.

Results page 4, last paragraph: What proportion of men responded correctly?

Discussion, line 174: You write that this is a trend, but you have only measured outcome at one timepoint, so you do not know anything about a trend. Maybe change the word “trend” to “results”.

Discussion, line 222: Reference 13 does not fit here. Please add appropriate reference.

Discussion, line 230: Reference (3) seems not to be added in a correct way with the reference manager. I also think you need a better reference for One Health.

Conclusion, line 289-290: You write that additional research is needed to explore antibiotic use in agriculture. This may be so, but this is not really something that comes out In your findings. Rather, you should state that more information about On Health and antibiotics in agriculture is needed to the public.

Author Response

Response to Reviewer 2

We appreciate Reviewer 2’s thoughtful comments and suggestions, which have helped us improve the clarity, structure, and focus of our manuscript. Below we address each comment in turn and indicate the corresponding line numbers where changes have been made in the tracked-changes document.

1. Introduction structure and content

We revised the Introduction by removing the methodological summary (previously lines 59–62) and discussion summary (previously lines 63–82). We also expanded the background with additional paragraphs to emphasize the importance of the study, including a clearer explanation of the One Health concept. Lines 49–88.

2. Participant Description in Results

The participant demographic section has been shortened for clarity. Detailed statistics are now presented in Table 1 to facilitate readability. Lines 92-97; Table 1.

3. Comparison to Kosovo Population

We included a comparison of our sample to the general Kosovo population in the second paragraph of the Discussion. Lines 181-187.

4. Legal Status of Antibiotic Sales

We clarified the legal framework for antibiotic prescription in Kosovo, noting that antibiotics should only be sold with a valid prescription. Lines 205-212.

5. Participant Recruitment and Sampling

We expanded the Methods section to better describe how participants were recruited, added details on sample size calculation, and noted the response rate. Lines 283-293.

We carefully addressed all minor corrections suggested by the reviewer. This included refining specific word choices for clarity in the abstract and results sections, correcting and updating references (including those related to One Health and rural healthcare access), clarifying gender-specific findings, and ensuring accurate wording to reflect the study's cross-sectional nature. All edits have been incorporated at the indicated line numbers throughout the manuscript.

We hope these revisions address all of Reviewer 2’s comments and suggestions. We thank the reviewer again for their valuable insights and support in improving our manuscript.

Sincerely,
The Authors

Reviewer 3 Report

Comments and Suggestions for Authors

Peer review report_1

The manuscript titled “Antibiotic Resistance Awareness in Kosovo: Insights from the WHO Antibiotic Resistance: Multi-Country Public Awareness Survey” presents a timely and relevant investigation into public awareness and behaviors regarding antibiotic resistance (AMR) in Kosovo, utilizing data from the WHO’s standardized multi-country survey. The topic is highly appropriate for the journal, addressing regional health behavior and global health threats through the lens of the One Health approach. Overall, the paper is clearly written, methodologically sound, and supported by appropriate literature. However, several areas require revision to enhance clarity.

Abstract:

  • Please consider including more specific numerical results (such as key percentages) in this section to better support the summary statements and give readers a clearer picture of the findings.

Introduction:

  • It may be helpful to clarify the rationale for focusing on Kosovo beyond the general mention of “lack of data.” Please consider briefly outlining any known challenges in antibiotic regulation or access to healthcare that make the setting particularly relevant.

Results:

  • Please consider reporting more detailed statistical outcomes, such as p-values and confidence intervals, to strengthen the interpretation of key group comparisons.

Discussion:

  • To improve this section, consider reducing repetition from the Results and offering more interpretative insights. For instance, what might be the real-world consequences of nearly 29% of respondents discontinuing antibiotics early?
  • Please elaborate on the sample's representativeness and discuss potential sampling bias, particularly in light of the urban and educational skew among respondents.

Conclusion:

  • Please consider adding concrete recommendations in this section, such as specific types of interventions, suggested policies, or how the findings could inform national strategies.

The article can be published after making the specified minor corrections.

Author Response

Response to Reviewer 3

We thank Reviewer 3 for their positive evaluation of our manuscript and for the insightful suggestions. Below, we provide our responses to each comment and outline where changes were made in the revised version.

Abstract: Include Specific Numerical Results

We revised the abstract to include key numerical findings that illustrate participants’ awareness and behaviors (e.g., 75% awareness of antibiotic resistance, 29% discontinued antibiotics early, 96% used prescriptions). Lines 20–29.

Introduction: Clarify Kosovo’s Relevance

We expanded the rationale for focusing on Kosovo by highlighting specific challenges in antibiotic regulation, enforcement, and healthcare access. These additions strengthen the case for why Kosovo is an important setting for this study. Lines 75-88.

Results: Include More Detailed Statistical Outcomes

We reported additional statistical details, including odds ratios, 95% confidence intervals, and p-values, to better support interpretation of group differences. Lines 112-150; see Table 2 too.

Discussion: Reduce Repetition and Add Interpretation

We reduced repetition of descriptive results and included further interpretation. For example, we elaborated on the implications of early discontinuation of antibiotics and connected this to healthcare system strain and AMR risk.

Discussion: Representativeness and Sampling Bias

We expanded the discussion on the sample’s limitations, particularly the urban and educational skew, and acknowledged how this may limit generalizability of the findings.
Lines 188-195.

Conclusion: Add Concrete Recommendations

We revised the conclusion to include actionable policy recommendations, such as implementing national AMR campaigns, improving prescription enforcement, and developing education initiatives aligned with WHO guidelines. Lines 331-344.

We are grateful for Reviewer 3’s thoughtful feedback, which has improved both the clarity and impact of our manuscript. All suggested changes have been incorporated in the revised version as outlined above.

Sincerely,
The Authors

Round 2

Reviewer 1 Report

Comments and Suggestions for Authors

Thank you for your thorough responses and revisions. The manuscript has adequately addressed the concerns raised.